# LLM-BS: Enhancing Large Language Models for Recommendation through Exogenous Behavior-Semantics Integration

## Abstract

Large language models (LLMs) are increasingly leveraged as foundational backbones in the development of advanced recommender systems, offering enhanced capabilities through their extensive knowledge and reasoning. Existing LLM-based recommender systems (RSs) often face challenges due to the significant differences between the linguistic semantics of pre-trained LLMs and the collaborative semantics essential for RSs. Typically, these systems apply pre-trained linguistic semantics while learning collaborative semantics from scratch using the LLM-Backbone. However, as LLM architectures are not inherently tailored for recommendation tasks, this approach results in inefficient learning of collaborative information, poor understanding of result correlations, and a failure to leverage traditional RSs features effectively. To address these challenges, we propose **LLM-BS**, a decoder-only **LLM**-based generative recommendation framework that integrates endogenous and exogenous **B**ehavioral and **S**emantic information in a non-intrusive manner. Specifically, we propose 1) a dual-source, knowledge-rich item indexing scheme that integrates indexing sequences for exogenous signals, enabling efficient link-wide processing; 2) a multi-scale reconfiguration alignment that non-intrusively guides the model toward a deeper understanding of both collaborative and semantic signals; 3) an Annealing Adapter designed to finely balance the model's recommendation performance with its comprehension capabilities. We demonstrate LLM-BS's effectiveness through rigorous testing on three public benchmarks.

## CCS Concepts

• **Information systems → Recommender systems**.

## Keywords

Recommender systems; large language models; Behavior-Semantic Collaboration

**ACM Reference Format:**
Anonymous Author(s). 2024. LLM-BS: Enhancing Large Language Models for Recommendation through Exogenous Behavior-Semantics Integration. In *Proceedings of ACM Web Conference 2025 (WWW '25)*. ACM, New York, NY, USA, 9 pages. https://doi.org/XXXXXXX.XXXXXXX

## 1 Introduction

Recommender systems (RSs) are tools designed to alleviate the phenomenon of information overload in Web environments by algorithmically analyzing user behavior to predict and push content that may be of interest to users. Typical recommender systems[11, 17, 20, 34] encode users and items as latent representations within a shared space to capture semantic similarities, followed by efficient retrieval using Approximate Nearest Neighbors (ANNs) algorithms[8, 16]. The distinct separation of these two phases often introduces performance limitations due to the absence of end-to-end optimization.

The emerging paradigm of recommender systems (RS) leveraging pre-trained large language models (LLMs) is showing great promise. Research across domains like vision[1, 23], speech[4, 15], and multimodality[42] demonstrates the broader applicability of LLMs, where various task instructions are encoded in language and fused with other forms, optimized via end-to-end training to adapt effectively to target domains. This paradigm harnesses the deeply embedded knowledge and logical reasoning capabilities of LLMs to discern intricate associations between user behavior and item semantics, leading to more accurate and nuanced recommendations.

Several recent studies investigate the potential roles of LLMs in recommender systems (RSs). Unlike traditional models that encode users and items as embedding vectors, some LLM-based RSs[2, 5, 13, 21] converts user behaviors and preferences, alongside the candidate item set, into discrete natural language text sequences or prompts. These prompts are then used to extract item-related information from the LLM's textual outputs. [7, 14, 24, 43] enhance collaboration by incorporating additional or existing tokens into the LLM to represent user and item IDs, which are then fine-tuned during specialized training to fit interaction data. [35, 38] employs exogenous collaboration models to obtain collaboration embeddings, which are integrated into the inputs of the LLM, thereby enriching the recommendation process.

But these paradigms suffer from several flaws:

**(1)** While the plain text approach can yield favorable outcomes in zero-shot recommendation[5, 13] , it primarily analyzes only the surface-level textual semantics of behavioral sequences. This method heavily relies on candidate sets and incurs significant computational overhead when modeling extensive historical sequences. **(2)** In real-world applications, where the number of candidate recommendation items vastly exceeds the vocabulary size of LLMs, the tokenization redundancy introduced by Vanilla IDs complicates LLMs' ability to accurately interpret commands. This redundancy results in low learning efficiency and a failure to effectively leverage semantic features. **(3)** The substantial disparity between the domains of external collaborative signals and the semantic signals of pre-trained LLMs means that directly integrating these signals can significantly disrupt the original functionalities of the LLMs.

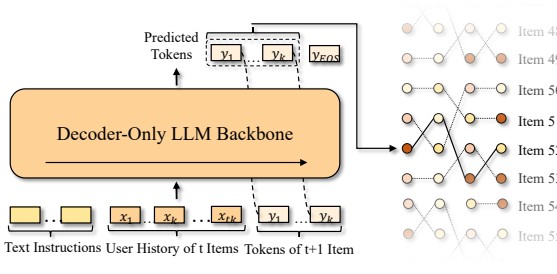

**Figure 1: A framework of LLM-Based recommendation. Converting recommendation tasks into Next Token generation tasks, directly generate target Items.**

Consequently, the model struggles to effectively process and interpret the information contained in these external signals[35, 38].

To overcome the challenges outlined, we introduce **LLM-BS**, a novel decoder-only LLM-based generative recommendation framework. In this framework, we compress massive exogenous signals into a few newly added tokens with extremely high compression ratios. Additionally, we incorporate a non-invasive multi-scale alignment reconstruction tasks and Multi-Stage Training that facilitates an efficient understanding and integration of exogenous behaviors, semantic signals, and recommendation data knowledge with the LLM's original parameters. Our approach is detailed through few key aspects:

Primarily, we introduce **Dual-source Knowledge-rich Item Indices** to address the inefficiencies of previous approaches that used atomic tokens to represent item IDs, which resulted in tokenization redundancy and overly discrete and independent semantics that did not effectively support the recommendation task. Our method efficiently characterizes large candidate sets with a minimal number of identifiers, incorporating a useful priori knowledge with a high compression ratio to integrate exogenous semantic and behavioral information into the decoding inference process. We implement an indexing structure where semantically similar items share identifier prefixes. Given the distinct domain differences between behavioral and semantic feature spaces, prior research in multimodal and bimodal models[3] has shown that even advanced encoder-side feature fusion approaches like Q-former[19] are insufficient for effective integration of dual-source features. Consequently, we discretize and separately splice the exogenous behavioral and semantic signals. This decoupled indexing scheme minimizes information loss from encoder-side feature fusion and enables the model to more effectively represent the complex interplay between behavior and semantics during subsequent training.

Furthermore, we have introduced **Non-Invasive Multiscale Alignment Reconstruction Tasks**. This approach helps the model process complex collaborative and semantic signals from tokens rich with exogenous information, while integrating the LLM's own parameters and reasoning capabilities to deepen its understanding of recommendations and associated tasks. Given the vast amount of exogenous semantic and behavioral signals compressed into a small number of tokens at a very high compression ratio, it is challenging

for the model to directly assimilate adequate exogenous knowledge. To address this, we have devised the Global Contrast Decompression Task and Comprehensive Interaction Modeling Tasks. These initiatives aid the model in decompressing extensive exogenous knowledge from a limited number of highly compressed tokens. By incorporating additional summarization tokens and leveraging the restricted context of recommendation data, these tasks effectively minimize the domain gap between natural language and collaborative semantics, enhancing the efficiency of the recommendation process. In addition, we introduced a multi-stage training scheme centered on the **Annealing Adapter**, which flexibly balances recommendation accuracy and model text inference capability.

The contribution of this paper can be concretely summarized as:

- We present LLM-BS, an innovative decoder-only LLM-based generative recommendation framework that synergistically integrates endogenous and exogenous behavioral and semantic information

- We propose a Dual-source Knowledge-rich Item Indices and a Multiscale Alignment Reconstruction Tasks that non-intrusively guides the model towards a deep understanding of collaborative and semantic signals. Additionally, we introduce an Annealing Adapter to optimize the balance between the model's textual reasoning abilities and recommendation accuracy.

- Extensive experiments across three public recommendation benchmarks demonstrate the superiority of LLM-BS over existing methods, emphasizing its effectiveness and robustness.

## 2 Related work

### 2.1 Sequential Recommendation

The use of deep sequential models for understanding user-item interactions in recommender systems has significantly evolved, with various approaches making notable contributions. GRU4REC[11] introduced the use of GRU-based RNNs to model sequential user behaviors effectively. SASRec[17] implemented self-attention mechanisms akin to those found in decoder-only transformer models to enhance recommendation accuracy. Drawing inspiration from the success of masked language modeling in NLP, BERT4Rec[30] applied transformers with masking techniques specifically tailored for sequential recommendation tasks. Additionally, TIGER[28] has started emphasizing the use of semantic IDs. In this approach, each item is represented by a series of tokens that reflect its related details, and the system predicts the sequence of upcoming item tokens using a seq2seq method. Additionally EAGER[36] advances the investigation by implementing a dual-stream generation architecture that incorporates both semantic and behavioral information. Recently, P5[7, 14] fine-tunes a pre-trained LLMs for multi-task recommender systems. In this study, we endeavor to further investigate a paradigm designed to mitigate the substantial discrepancies between LLMs in recommendation tasks and their original training tasks by integrating exogenous semantic and behavioral information.

### 2.2 LMs for Recommendation

Recently, LLMs have been utilized in recommendation tasks due to their ability to understand, generate, and infer natural language properties. LLM-based RSs[24] constructs user/item correlations

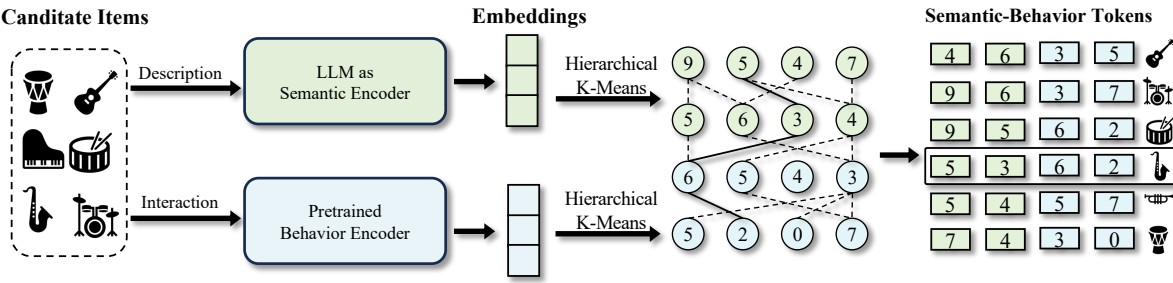

**Figure 2: The illustration of Dual-source Knowledge-rich Item Indice. Incorporate massive exogenous behavioral semantic signals into project indices with extremely high compression ratios.**

through its powerful high-quality textual representations and extensive external knowledge, and is expected to solve the problems of poor generalization[22] and poor performance of traditional RSs on sparse historical interaction data, etc. Chat-Rec[6] aims to enhance conversational recommendation systems by integrating ChatGPT's interactive capabilities with established recommendation models, such as MF[18] and LightGCN[10]. P5[7] fine-tunes a pre-trained large language model for multi-task recommender systems, utilizing the LLM tokenizer (SentencePiece tokenizer) to generate tokens from randomly assigned item pseudo-IDs. M6[5]explores the use of item text information (such as names) as identifiers for items. LC-Rec[39] designs a learning-based vector quantization method to generate ID from Item's semantic representation and proposes alignment tuning tasks to enhance the intergration of collaborative semantics in LLMs. However, merely using directive-based fine-tuning falls short in effectively leveraging the inherent capabilities of LLMs to understand collaborative information and inadequately learn from the implicit interaction data crucial for recommendations. Recently, new research has emerged to bridge the significant gap between pre-trained language models and recommendation tasks. CoLLM[38] infuses behavior information into LLMs by incorporating representations from an external collaborative model into the input. In this work, we aim to further explore recommender frameworks that can integrate endogenous and exogenous behavioral and semantic signals based on LLM.

## 3 METHODOLOGY

### 3.1 Problem Formulation

Sequential recommendation is a crucial metric in LLM-based recommender systems (RSs). We transform the traditional two-tower model, which computes similarity followed by reordering, into a generative recommendation paradigm. In this framework, each item $\mathbf{x}$ is represented by a set of tokens $\mathbf{Y} = [\mathbf{y_1}, \mathbf{y_2}, \cdots, \mathbf{y_k}] \in \mathcal{Y}$. As illustrated in 3, given an input sequence $\mathbf{X}$, which includes instructions and the interaction history, the sequence of the target item $\mathbf{Y}$ is generated directly in an autoregressive manner. The probability can be calculated by:

$$p(\mathbf{Y}|\mathbf{X}) = \prod_{i=1}^{k} p(\mathbf{y}_i|\mathbf{X}, \mathbf{y_1}, \mathbf{y_2}, \ldots, \mathbf{y_{i-1}}) \tag{1}$$

### 3.2 Dual-source Knowledge-rich Item Indices

Some existing LLM-based methods utilize bracket notations like "<item_i>" as newly-introduced atomic tokens to represent items. However, this method can be problematic in data-rich real-world scenarios, where the number of potential recommended items greatly exceeds the vocabulary of LLMs. This leads to tokenization redundancy, making it challenging for LLMs to process commands accurately. Moreover, the description-based approach[2], which assigns tokens to index items based on the semantics of item titles or descriptions, introduces a strong inductive bias. This can obscure the true intent of user behaviors, as it does not model behavioral sequences clearly and unbiasedly, compromising the model's ability to understand and predict user preferences effectively.

Additionally, existing methods often overlook the value of exogenous prior knowledge. Further, our experiments show that using either exogenous behavioral or semantic signals in isolation does not outperform random sampling. Remarkable results are only achieved when these two signals are effectively integrated, demonstrating subtle interaction, understanding, and cooperation.

To address our objectives, we aim to: 1) introduce a minimal number of tokens to efficiently represent a vast set of candidates; 2) infuse useful a priori knowledge into identifiers to incorporate exogenous semantic and behavioral information about items into the reasoning process; and 3) design an indexing structure where semantically similar items share identifier prefixes. To achieve these goals, we utilize a discretized indexing algorithm that encodes dual-source information for item representation. As illustrated in 2, for any given item along with its descriptive text (title, synopsis, etc.), semantic embeddings are derived using pre-trained language models (e.g., T5[27], Llama[32]). In this study, we specifically employ the LLM-Backbone itself for semantic extraction:

$$\text{hiddenstate}_t = llm(x_t, \text{hiddenstate}_{t-1}), \quad t = 1, 2, 3, ..., n$$
$$z_i^s = \frac{1}{n} \sum_{i=1}^{n} \text{hiddenstate}_i \tag{2}$$

The descriptive text $x_{1:n}$ of item $i$ is entered sequentially into the last hidden state transformed by the LLM and averaged as the semantic representation $z^s$ of the item. Behavioral features $z^b$ are extracted by the encoder of a two-tower model (e.g., DIN[40]) that uses only ID sequences as recommendations:

$$z_i^b = BehaviorEncoder(i) \tag{3}$$

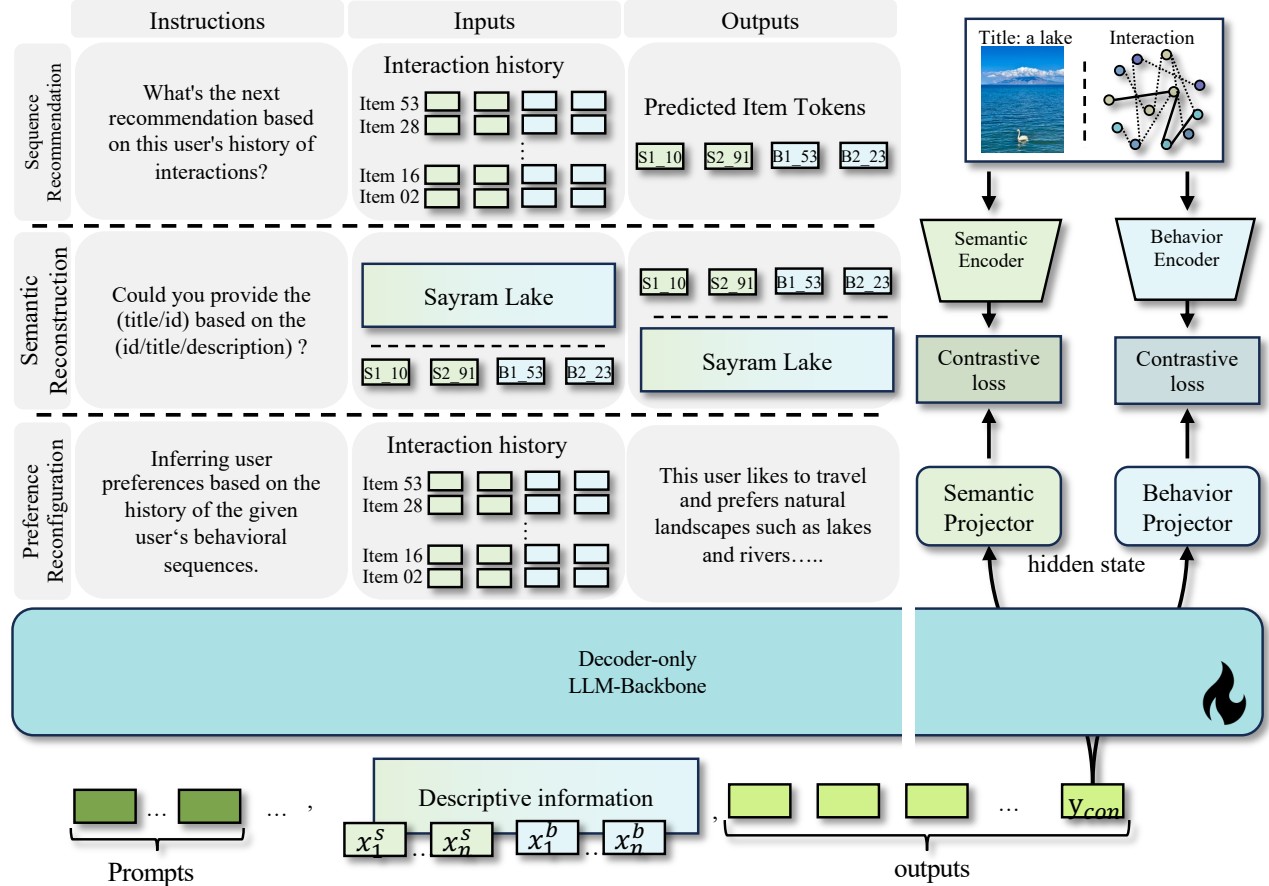

**Figure 3: Overview of Non-Invasive Multiscale Alignment Reconstruction Tasks. Includes Global Contrast Decompression Task (GCT), Comprehensive Interaction Modeling Task (CIT). These tasks facilitate the LLM-Backbone's comprehension of complex exogenous signals from these densely packed, knowledge-rich tokens in a non-invasive manner, and to integrate the llm's reasoning capabilities for deep understanding of recommendations.**

Given the domain disparities between behavioral and semantic feature spaces, prior research has shown that even advanced encoder-side feature fusion techniques (e.g., Q-former[19]) often result in significant compression losses and fail to effectively integrate dual-source features. This places supernumerary strain on the decoding process. Consequently, we opt to separately and discretely process the exogenous semantic and behavioral signals. While vector quantization is commonly used for discretization, it proves unstable for training, leading to issues like item ID conflicts.

To facilitate reproducibility in our study, we employ hierarchical K-Means to discretize the semantic embedding $Z^s$ and behavioral embedding $Z^b$, where each cluster is progressively subdivided into $k$ child clusters until each cluster contains only a single item. The embeddings for each item are discretized into $C^s$ and $C^b$. Specifically, the j-th ID of the i-th item is denoted as $c_{ij}^t$, where t includes s for semantic, b for behavioral, and u for the unitive. Theoretically, with each item represented by four tokens and each token capable of 256 distinct values, this method can uniquely characterize

up to $256^4 = 4294967296$ items. This capacity is more than adequate for real-world applications, ensuring efficiency in vocabulary expansion and the subsequent encoding and decoding processes.

## 3.3 Non-Invasive Multiscale Alignment Reconstruction Tasks

*3.3.1 **Global Contrast Decompression Task**.* After incorporating additional tokens to represent items, we achieve a high compression ratio ($\approx 2{,}000{,}000{:}1$), which significantly condenses massive exogenous signals into a very small number of tokens. This extreme compression ratio presents a challenge for the model to independently learn substantial, useful knowledge. Drawing inspiration from[36, 39], we have devised a series of multi-scale alignment reconstruction tasks. These tasks facilitate the LLM's comprehension of complex collaborative and semantic signals from these densely packed, knowledge-rich tokens in a non-invasive manner, and to integrate the llm's own parameters and reasoning capabilities for deep understanding of recommendations and their related tasks.

We introduce the Global Contrast Decompression Task (GCT), a method that non-intrusively enhances the model's ability to quickly and easily interpret knowledge-rich indices at extreme compression rates. This is achieved by incorporating additional summarization token and trainable Decompression Guidance Projectors.

$$\text{seq} = \{X^{\text{Prompts}}, X_1^u, ......, X_n^u, y_1^s, ......, y_k^b, y_{[CON]}\} \quad (4)$$

Where $X^{\text{Prompts}}$ denotes the sequence of text prompts $\{x_1^P, ..., x_m^P\}$ and $X_i^u$ represents the indexed tokens for the i-th item, reflecting the user's chronological behavior sequence. As outlined in 3.2, $y_j^t$ denotes the j-th level of the predicted item tokens, where $t = b$ corresponds to the item's behavioral token, and $t = s$ to the semantic token. The summary character $y_{[CON]}$ is strategically placed at the end to encapsulate the global knowledge of the preceding sequence.

To efficiently transmit exogenous dual-source signals into the preordered tokens through gradient updating, we introduce non-intrusive Decompression Guidance Projectors $f^t$. This projector transforms the global hidden state distilled by $y_{[CON]}$ into semantically and behaviorally-guided latent states. Additionally, we employ a contrastive learning paradigm that utilizes original exogenous semantic embbeddings $Z^s$ and behavioral embbeddings $Z^b$ to accelerate and assist the decompression process of hyper-compressed Tokens.

$$\text{hiddenstate}_t = \hat{llm}(x_t, \text{hiddenstate}_{t-1}), \quad t = 1, 2, 3, ..., n$$
$$\mathcal{L}_{\text{con}}^t = \mathcal{F}(f^t(\text{hiddenstate}_{[CON]}), Z^t), \quad t \in \{b, s\} \quad (5)$$

The total contrastive loss $\mathcal{L}_{\text{con}}$, is calculated by proportionally summing $\mathcal{L}_{\text{con}}^t$ and $\mathcal{L}_{\text{con}}^b$. The function $\mathcal{F}(\cdot, \cdot)$ serves as the metric for contrastive learning. Importantly, the Decompression Guidance Projectors $f^t$ are utilized only during training, not in inference.

*3.3.2* ***Comprehensive Interaction Modeling Task***. To effectively harness the inference capabilities, pre-training knowledge, and trainable parameters of the LLM-Backbone for fitting recommendation data, we have restructured the traditional sequence recommendation task and its auxiliary tasks into a Next Token Prediction task, which LLMs are good at. Unlike using additional selectors as suggested by[43], we contend that this could alter the model's output form and output domain distribution, potentially compromising the original capabilities of the LLM-Backbone and diminishing the framework's generalizability across different backbones.

As illustrated in 3, Comprehensive Interaction Modeling is segmented into three subtasks: "Sequence Recommendation Task," "Semantic Reconstruction Task," and "Preference Reconfiguration Task." These tasks effectively leverage the model's own parameters to integrate exogenous signals, recommendation data knowledge, and the model's intrinsic reasoning capabilities organically.

## 3.4 Initial training, Annealing Adapter Tuning and Inference

*3.4.1* ***Initial training***. In the initialization phase of enhancing the model's recommendation capabilities, we devised various conditional language modeling objectives. This strategy encourages

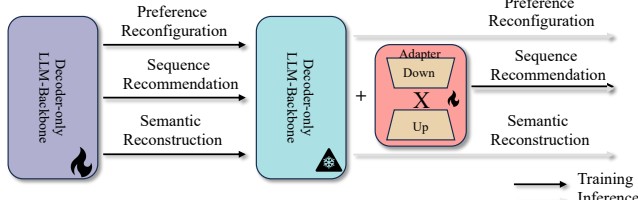

**Figure 4: The illustration of multi-stage training & inference process.**

highly divergent models, compared to pre-recommendation pre-train tasks, to cultivate in-depth generalization, understanding, and reasoning abilities pertinent for recommendation tasks.

The initial training can be formulated as follows:

$$\max_{\Phi} \sum_{(x,y)\in\mathcal{Z}} \sum_{t=1}^{|y|} \log \left( P_{\Phi+\varphi_r} (y_t \mid x, y_{<t}) \right) \quad (6)$$

$x$ represents the "Instruction Input". $y$ denotes the "Instruction Output" within the initial training data. $y_t$ stands for the $t$-th token of $y$. $\Phi$ corresponds to the original parameters of the LLM-Backbone. $\varphi_r$ represents the additional parameters in Sequence Recommendation Task (SRT), and $\mathcal{Z}$ refers to the training set. We combine the generation and exogenous Semantic, Behavioral Reconstruction Loss to train our model, given by:

$$\mathcal{L} = \mathcal{L}_{gen} + I_{\text{SRT}}(\lambda_1 \mathcal{L}_{con}^s + \lambda_2 \mathcal{L}_{con}^b) \quad (7)$$

Where $I_{\text{SRT}}$ is an indicator function that is 1 if the task is SRT and 0 otherwise. $\lambda_1$ and $\lambda_2$ are loss coefficients.

*3.4.2* ***Annealing Adapter Tuning***. we observed that annealing with restricted quantities of high-grade sequence recommendation data considerable improves the performance of the LLM-Backbone on pivotal benchmarks subsequent to the initial training of recommendation capabilities.

Achieving the optimal solution for enhancing sequence recommendation performance remains a formidable challenge without adjusting the data volume ratio across various tasks. Integrating tasks such as sequence recommendation, preference reconfiguration, and semantic reconstruction, while neglecting to bridge the gap between natural language processing and sequential behavior, complicates the optimization of sequence recommendation performance without modifying the proportion of data volume allocated to different tasks.

Conversely, training on a limited set of high-grade sequence recommendation tasks during the Annealing Training phase can also impair the model's original capabilities due to the significant disparity between the language semantics modeled by LLMs and the collaborative semantics implicit in recommender systems. Therefore, the use of an Adapter to introduce additional parameters in this phase, as shown in 4, constitutes an efficient and pragmatic approach to mitigate the adverse effects associated with Annealing Training.

Formally,

**Table 1: Statistics of the Datasets.**

| Dataset | #Users | #Items | #Interactions | #Sparsity |
|---|---|---|---|---|
| Beauty | 22,363 | 12,101 | 198,360 | 0.00073 |
| Sports and Outdoors | 35,598 | 18,357 | 296,175 | 0.00045 |
| Instruments | 24,733 | 9,923 | 206,153 | 0.00083 |

$$\max_{\varphi_a} \sum_{(x,y) \in \mathcal{Z}} \sum_{t=1}^{|y|} \log \left( P_{\Phi + \varphi_r + \varphi_a} \left( y_t \mid x, y_{<t} \right) \right) \tag{8}$$

where $\varphi_a$ denotes the parameters of Annealing Adapter.

*3.4.3* **Inference**. It is noteworthy that the additional parameter $\phi$, brought forth by the SRT, is disregarded during the inference stage. Moreover, employing the Annealing Adapter dynamically to meet varying task demands acts as a potent strategy for achieving a flexible balance between the model's textual reasoning abilities and recommendation accuracy.

# 4 EMPIRICAL STUDY

We analyze the proposed LLM-BS method on three datasets and demonstrate its effectiveness by answering the following research questions:

- RQ1: How does LLM-BS compare to state-of-the-art sequential recommendation (traditional, LLM-based) methods in different datasets?
- RQ2: How do the components of LLM-BS (e.g., Dual-source Knowledge-rich Item Indices, Non-invasive Contrast Task, Annealing Adapter ) affect the performance?
- RQ3: How do various ablation variants and hyper-parameter adjustments impact the performance of LLM-BS?

## 4.1 Experimental Setting

*4.1.1* **Dataset**. We conducted experiments using three real-world public datasets of Amazon product reviews[9, 26], which are among the most widely utilized benchmarks for sequence recommendation. Specifically, the experiments focused on three subcategories: "Beauty", "Sports and Outdoors" and "Musical Instruments". These categories include user reviews and item metadata spanning from May 1996 to July 2018. In line with previous studies[12, 29, 37], we utilized the 5-core dataset approach, which excludes unpopular items and inactive users with fewer than five interaction records. The statistics for these datasets are presented in 1.

*4.1.2* **Evaluation Metrics**. We utilize two widely recognized criteria for the matching phase: Recall and Normalized Discounted Cumulative Gain (NDCG). We present metrics calculated for the top 5/10 recommended candidates. In line with the standard evaluation protocol[17], we adopt the leave-one-out method for assessments. Specifically, for each sequence of user behavior, the most recent item is designated as the test data, the next most recent as the validation data, and all previous interactions are used for training. During training phases, we restrict the user's historical item count to 20. Additionally, for generative methods employing beam search, we consistently set the beam size to 20.

*4.1.3* **Implementation Details**. We utilize Llama-7b[32] as LLM-Backbone. In constructing the item indexes, LLM-Backbone itself and DIN[40] as encoders, combining semantic and behavioral indexes to form each item's final ID. For training, our approach mirrors that of LC-Rec for ease of comparison, employing the AdamW optimizer with a learning rate set to 5e-5 and weight decay at 0.01. We use a cosine scheduler with warmup to adjust the learning rate effectively. We implement data parallelism and gradient accumulation to achieve an overall batch size of 128. For GCT, we adopt InfoNCE to serve as the loss metric.

## 4.2 Performance Comparison (RQ1)

*4.2.1* **Baselines**. To demonstrate the superiority of all our methods, we compare the following five categories of methods:
(1) *Traditional seqential methods*

- **GRU4REC** [11]: An RNN-based sequential recommendation model that utilizes GRU model to encode the item sequence.
- **Caser** [31]: a CNN-based approach that utilizes horizontal and vertical convolutional layers to model the patterns in user behavior.
- **HGN** ] [25]: employs hierarchical gating networks to effectively discern long-term and short-term user preferences.

(2) For *transformer-based methods*, we have:

- **S^3-Rec** [41]: S^3-Rec enhances sequential recommendation by pre-training a bidirectional Transformer using self-supervised learning tasks, focusing on maximizing mutual information.
- **BERT4Rec** [30]: Utilizes a bidirectional Transformer to overcome the constraints of unidirectional models..
- **FDSA** [37]: models feature sequence transition patterns using a self-attention module.

(3) For *generative methods*, we have:

- **TIGER** [28]: TIGER employs T5 to generate semantic IDs for items and uses an autoregressive decoding process to identify target candidates.
- **P5-CID** [14]: leverages collaborative signals to construct ID identifiers for T5-based generative recommender model.

(4) For *LLM-Based methods*, we have:

- **LC-Rec** [39]: LC-Rec designs a vector quantization method to generate semantic IDs and use Llama as backbone to autoregressively decodes the identifiers of the target candidates items.
- **LETTER** [35]: Integrates collaborative signals into LLM-Backbone through a series of regularizations.

*4.2.2* **Overall Performance**. We provide a detailed report in 2 on the sequence recommendation performance of our method across three datasets, comparing it against various baseline models. Specifically for the Instruments dataset, we used the official LC-Rec checkpoints to rerun the inference with the conflicts removed. The results lead to several key observations :

LLM-BS obtains better results than base modes on all three datasets. We believe that this is mainly attributed to the fact that LLM-BS effectively introduces exogenous semantic and behavioral signals and makes it possible for the LLM-Backbone to understand this information in depth through a series of non-invasive tasks.

**Traditional baselines** employ a simple inner-product matching approach, which segments the process and limits its ability to

**Table 2: Performance comparison of different methods. The best performance is highlighted in bold while the second best performance is underlined. The last column indicates the improvements over the best baseline models and all the results of LLM-BS are statistically significant with p < 0.05 compared to the best baseline models.**

| Dataset | Metric | Traditional | | | Transformer-based | | | Generative | | LLM-based | | Improv. |
|---|---|---|---|---|---|---|---|---|---|---|---|---|
| | | GRU4REC | Caser | HGN | Bert4Rec | S^3-Rec | FDSA | P5-CID | TIGER | LC-Rec | Ours | |
| Beauty | Recall@5 | 0.0164 | 0.0205 | 0.0325 | 0.0203 | 0.0387 | 0.0267 | 0.0400 | 0.0454 | 0.0482 | **0.0548** | 13.69% |
| | Recall@10 | 0.0283 | 0.0347 | 0.0512 | 0.0347 | 0.0647 | 0.0407 | 0.0590 | 0.0648 | 0.0681 | **0.0830** | 21.88% |
| | NDCG@5 | 0.0099 | 0.0131 | 0.0206 | 0.0124 | 0.0244 | 0.0163 | 0.0274 | 0.0321 | 0.0327 | **0.0369** | 12.84% |
| | NDCG@10 | 0.0137 | 0.0176 | 0.0266 | 0.0170 | 0.0327 | 0.0208 | 0.0335 | 0.0384 | 0.0409 | **0.0459** | 12.22% |
| Sports | Recall@5 | 0.0129 | 0.0116 | 0.0189 | 0.0115 | 0.0251 | 0.0182 | 0.0313 | 0.0264 | 0.0304 | **0.0373** | 19.17% |
| | Recall@10 | 0.0204 | 0.0194 | 0.0313 | 0.0191 | 0.0385 | 0.0288 | 0.0431 | 0.0400 | 0.0451 | **0.0569** | 26.16% |
| | NDCG@5 | 0.0086 | 0.0072 | 0.0120 | 0.0075 | 0.0161 | 0.0122 | 0.0224 | 0.0181 | 0.0196 | **0.0251** | 12.05% |
| | NDCG@10 | 0.0110 | 0.0097 | 0.0159 | 0.0099 | 0.0204 | 0.0156 | 0.0262 | 0.0225 | 0.0246 | **0.0315** | 20.23% |
| Instruments | Recall@5 | 0.0821 | 0.0543 | 0.0813 | 0.0671 | 0.0863 | 0.0834 | 0.0827 | 0.0863 | 0.0964 | **0.0991** | 2.80% |
| | Recall@10 | 0.1031 | 0.0710 | 0.1048 | 0.0822 | 0.1136 | 0.1046 | 0.1016 | 0.1064 | 0.1177 | **0.1224** | 3.99% |
| | NDCG@5 | 0.0698 | 0.0355 | 0.0668 | 0.0560 | 0.0626 | 0.0681 | 0.0708 | 0.0738 | 0.0819 | **0.0851** | 3.91% |
| | NDCG@10 | 0.0765 | 0.0409 | 0.0744 | 0.0608 | 0.0714 | 0.0750 | 0.0768 | 0.0803 | 0.0890 | **0.0926** | 4.04% |

**Table 3: Performance comparison of LETTER and our method. For fair comparison, llm-bacbone is uniformly Llama2-7b. All the results of LLM-BS are statistically significant with p < 0.05.**

| Model | Instruments | | | |
|---|---|---|---|---|
| | R@5 | R@10 | N@5 | N@10 |
| TIGER | 0.0870 | 0.1058 | 0.0737 | 0.0797 |
| LETTER-TIGER | 0.0909 | 0.1122 | 0.0763 | 0.0831 |
| LC-Rec | 0.0824 | 0.1006 | 0.0712 | 0.0712 |
| LETTER-LC-Rec | 0.0913 | 0.1115 | 0.0789 | 0.0854 |
| LLM-BS (Llama2-7B) | **0.0994** | **0.1206** | **0.0854** | **0.0922** |
| Improv. | 8.95% | 7.48% | 8.26% | 8.02% |

effectively model complex user interaction histories and intentions. Moreover, this approach's computational complexity grows exponentially with the candidate set, also restricting the representational space size. In contrast, LLM-BS aligns with the generative recommendation paradigm. It not only leverages pre-training knowledge to enhance recommendation-relevant capabilities, but it also reduces computational costs by directly generating the target item ID through beam search. This approach expands the limitations of latent space size in item representation, allowing it to incorporate significantly more exogenous information.

**Generative recommendation, llm-based approaches** (TIGER, LC-REC etc.) neglected the importance of exogenous behavioral signals for sequence recommendation. While the transformer architecture with generation loss works well in various domains, it is not designed for the task of sequence recommendation. These non-native approaches ignore the rank-order relationship of the candidates in the recommendation task, which leads to poor model performance on ranking-related metrics such as ndcg. Therefore,

we believe that introducing additional behavioral signals is the key to improving the overall performance of model recommendation without changing the model architecture and training process.

There are also some approaches that attempt to **incorporate exogenous behavioral signals into the recommendations** (P5-CID, LETTER). LETTER, for instance, integrates collaborative signals into discrete coding through a series of regularizations. However LETTER does not have open source code and is only implemented as Llama2-7b[33], we evaluated our LLM-BS using the Llama2-7b on the Instruments dataset, as detailed in 3. Our method outperforms LETTER by over 8% across all metrics. We contend that even sophisticated encoder-side feature fusion methods can introduce additional compression loss, hindering the efficient integration of multi-source features. Therefore, allowing the LLM-Backbone itself to handle the fusion of information without introducing extra generalization bias at the input emerges as a simpler and more effective strategy.

## 4.3 Ablation Study (RQ2)

In the ablation experiments, the Sequence Recommendation Prediction Task was used as the core metric to evaluate the performance impact of each component. The main components of LLM-BS include Dual-source Knowledge-rich Item Indices (DKI), Global Contrast Decompression Task (GCT), and Annealing Adapter Tuning (AAT). The results are reported in 4, we can observe that:

- Removing LLM-BS of the DKI, GCT, ATT (random index) achieves the worst results in different datasets, but still outperforms the vast majority of traditional baselines. This underscores the inherent superiority and robustness of our foundational framework in addressing the sequence recommendation task, and highlights significant potential for further development and enhancement in future work.

**Table 4: Ablation studies by selectively discarding the Dual-source Knowledge-rich Item Indices (DKI), Global Contrastive Task (GCT), and Annealing Adapter Tuning (AAT).**

| Variants | | | Beauty | | | | | Musical Instruments | | | | |
|---|---|---|---|---|---|---|---|---|---|---|---|---|
| DKI | GCT | AAT | R@1 | R@5 | R@10 | NDCG@5 | NDCG@10 | R@1 | R@5 | R@10 | NDCG@5 | NDCG@10 |
| | | | 0.0135 | 0.0453 | 0.0650 | 0.0295 | 0.0358 | 0.0631 | 0.0883 | 0.1071 | 0.0757 | 0.0817 |
| ✓ | | | 0.0152 | 0.0499 | 0.0760 | 0.0329 | 0.0413 | 0.0696 | 0.0978 | 0.1199 | 0.0802 | 0.0886 |
| ✓ | ✓ | | 0.0175 | 0.0513 | 0.0781 | 0.0346 | 0.0432 | 0.0694 | 0.0981 | 0.1215 | 0.0839 | 0.0914 |
| ✓ | ✓ | ✓ | **0.0176** | **0.0544** | **0.0817** | **0.0363** | **0.0451** | **0.0707** | **0.0991** | **0.1225** | **0.0852** | **0.0927** |

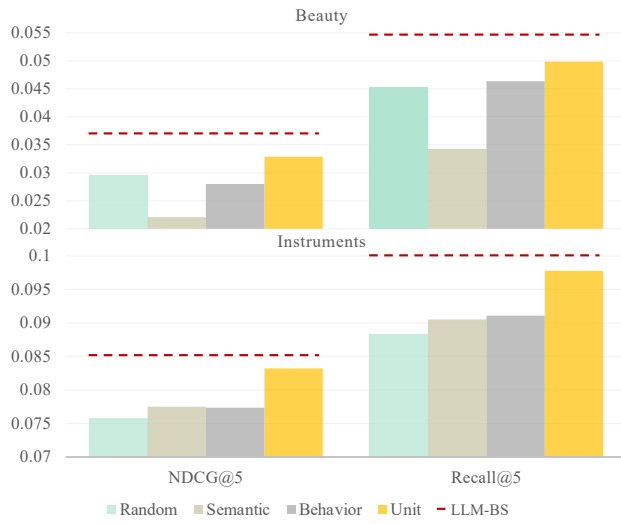

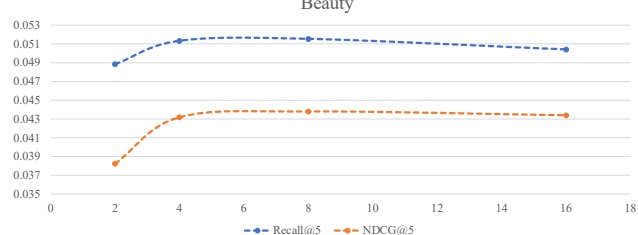

**Figure 6: Analysis of the performance impact of Items Indices schemes of different lengths.**

**Figure 5: The performance of our architecture (w/o GCT, AAT), under indexing with different exogenous signal compositions.**

- Removing DKI significantly impacts sequence recommendation performance, illustrating the base model's effectiveness in enhancing recommendations by integrating exogenous behavioral and semantic information. This also showcases DKI's capability to encapsulate vast information within a few tokens at a high compression ratio.
- GCT significantly enhances the NDCG metrics compared to Recall, indicating its efficacy in decoding exogenous behavioral signals compressed by DKI. GCT effectively incorporates external knowledge that optimizes the model's ability to sequence recommendations. This underscores our architecture's adaptability in harnessing distinct properties from various exogenous information sources.

## 4.4 Further Analysis (RQ3)

As depicted in 5, we conducted performance experiments on the Beauty and Instruments datasets using various exogenous signal indexing methods. We tested four indexing strategies: **(1)** Random, where each level of indices is randomly selected from candidates and ensured to be conflict-free; **(2)** Semantic, utilizing indices derived

solely from exogenous textual semantic signals via a discretization algorithm; **(3)** Behavior, using indices generated solely from exogenous behavioral signals via a discretization algorithm; and **(4)** Unit, combining indices from both Semantic and Behavior. The Unit index significantly exceeds the sum of the individual contributions from the two sources, yielding much higher results.

**To our shock**, in the Beauty dataset, the Semantic index performs worse than the Random index, likely due to a high compression ratio that complicates the model's ability to decode separate exogenous signals, especially after removing GCT and AAT. This often results in a diminished or even negative impact on recommendation performance. More intriguingly, the integration of exogenous Behavioral signals enables effective interaction between the dual information streams, enhancing their mutual comprehension and decoding. This synergy not only mitigates the negative impacts but also transforms them into substantial positive outcomes.

In 6, we demonstrate the impact of varying lengths of the Item Indices scheme on performance within the Beauty dataset. Observations indicate that four layers of Indices provide sufficient information for effective model learning. Larger layers does not complicate the generation of legitimate IDs for the model. however, it does result in increased inference times.

## 5 CONCLUSION

In this paper, we introduce LLM-BS, a novel decoder-only, LLM-based generative recommendation framework that seamlessly integrates both endogenous and exogenous behavioral and semantic information non-intrusively. Extensive experiments validate the effectiveness and robustness of LLM-BS, showcasing superior performance compared to existing state-of-the-art methods.

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
