# OpenReview forum: "LLM-BS: Enhancing Large Language Models for Recommendation through Exogenous Behavior-Semantics Integration"
_ACM.org/TheWebConf/2025/Conference — WWW 2025 Poster_

### Official Review · Reviewer_xahL · 2024-11-09

**Novelty:** 4
**Technical Quality:** 5

**Review:**

This paper introduces LLM-BS, a decoder-only LLM-based generative recommendation framework. In this framework,  massive signals are compressed into a few newly added tokens with extremely high compression ratios. A non-invasive multi-scale alignment reconstruction tasks is introduced and multi-stage training is developed to facilitate an efficient understanding and integration of exogenous behaviors,
semantic signals, and recommendation data knowledge with the LLM’s original parameters. Additionally, an annealing adapter is introduced to optimize the balance between the model’s textual reasoning abilities and recommendation accuracy. Experiments on three datasets show performance improvement.

Strength
1. Presentation is clear.
2. The research topic on LLM-based generative recommendation is timely and of interest to the community.
3. The proposed method shows performance improvement.

Weakness
1. No hyperparameter sensitivity analysis (e.g., study the effect of different combinations of $\lambda_1$ and $\lambda_2$).
2. Novelty is limited. Hierarchical clustering for item indexing is studied in [1]. Semantic index reconstruction is studied in [2], adapter tuning for LLM-based recommendation is studies in [3], and there are already a lot of LLaMa-based recommender systems. In this perspective, this paper is just a simple combination of existing techniques.
3. The proposed approach might have scalability issue (see the questions below).

References:

[1] Tay, Yi, Vinh Tran, Mostafa Dehghani, Jianmo Ni, Dara Bahri, Harsh Mehta, Zhen Qin et al. "Transformer memory as a differentiable search index." Advances in Neural Information Processing Systems 35 (2022): 21831-21843.

[2] Sun, Weiwei, Lingyong Yan, Zheng Chen, Shuaiqiang Wang, Haichao Zhu, Pengjie Ren, Zhumin Chen, Dawei Yin, Maarten Rijke, and Zhaochun Ren. "Learning to tokenize for generative retrieval." Advances in Neural Information Processing Systems 36 (2024).

[3] Fu, Junchen, Fajie Yuan, Yu Song, Zheng Yuan, Mingyue Cheng, Shenghui Cheng, Jiaqi Zhang, Jie Wang, and Yunzhu Pan. "Exploring adapter-based transfer learning for recommender systems: Empirical studies and practical insights." In Proceedings of the 17th ACM International Conference on Web Search and Data Mining, pp. 208-217. 2024.

**Questions:**

This paper chooses a very high compression ratio to derive semantic indices. Under such a high ratio, information loss is very likely, and it is possible that even with the designed reconstruction task, the recommender system still struggles to learn meaningful patterns from such a high-ratio compression. Also, this approach will result in higher information loss when the number of items get larger, which may not be scalable in real-world settings. If possible, could you please provide additional experiments on larger dataset to demonstrate whether the proposed approach is effective?

**Reviewer Confidence:**

3: The reviewer is confident but not certain that the evaluation is correct

**Scope:**

4: The work is relevant to the Web and to the track, and is of broad interest to the community

---

### Official Review · Reviewer_XMJM · 2024-12-01

**Novelty:** 5
**Technical Quality:** 3

**Review:**

## Review
This is an interesting manuscript with innovative ideas and an ambitious framwork. In general,
### Pros
- The paper proposes a novel framework for integrating semantic and behavioral signals into LLM-based recommender systems. The framework, LLM-BS, introduces and combines innovative methods to address critical gaps in current systems.
-  The motivation behind the work is clearly explained, and the manuscript outlines the objectives and empirical results effectively.
-  The framework is extensively tested across multiple datasets using standard evaluation metrics. The use of ablation studies strengthens the claims made in the paper.
-  By tackling tokenization redundancy and improving the integration of behavioral signals, the framework has the potential to bridge the gap between LLMs and collaborative recommendation tasks.
However,
### Cons
-  The methodology section is overly complex and hard to follow. Technical components such as the Global Contrast Decompression Task (GCT), Comprehensive Interaction Modeling Task, and Annealing Adapters lack deeper explanations and intuitive examples. Additionally, the workflow from embedding extraction to tokenization, compression, and decoding remains abstract without a clear step-by-step explanation.
 - The claim that integrating semantic and behavioral signals leads to superior performance and that either one in isolation does not outperform random sampling lacks empirical evidence. For instance, the statement on Page 1:
  > "Using either exogenous behavioral or semantic signals in isolation does not outperform random sampling"
  is not supported with any data or experiments.
-   Adding multiple components (e.g., dual indices, GCT, projectors, multi-scale alignment) raises concerns about the practicality of the framework due to high computational overhead. There is no discussion about whether simpler, more efficient methods could achieve comparable results.
- Semantic and behavioral signals are processed independently, and later combined. By discretizing both signals into separate clusters, cross-signal relationships are ignored. This contradicts the claim on Page 2:
  > "This decoupled indexing scheme minimizes information loss from encoder-side feature fusion and enables the model to more effectively represent the complex interplay between behavior and semantics."
- The claim of achieving efficiency and scalability on Page 4 is contradicted by the use of computationally expensive tasks, such as hierarchical K-Means and projector-based decompression, and no quantitative analysis of computational burden is provided:
  > "This capacity is more than adequate for real-world applications, ensuring efficiency in vocabulary expansion and the subsequent encoding and decoding processes."
-   The paper emphasizes on Page 4 achieving a high compression ratio (~2,000,000:1), yet fails to clearly justify why such an extreme ratio is necessary or how it is achieved in the paper.
- The Global Contrast Decompression Task (GCT) introduces trainable projectors, which modify the LLM backbone. This contradicts the claim that GCT is "non-invasive" on Page 5. A clearer definition and expalaination of "non-invasive" could be beneficial and aviod confusion:
  > "This is achieved by incorporating additional summarization tokens and trainable Decompression Guidance Projectors."
- The Annealing Adapter is said to improve performance by balancing general reasoning and recommendation-specific tasks, but there is no logical explanation or empirical evidence for this claim. Page 5:
  > "Annealing with restricted quantities of high-grade sequence recommendation data considerably improves the performance of the LLM-Backbone."
- The mathematical notations throughout the paper are inconsistent. This could create confusion about the structure and flow of the methodology. For example on Page 3:
  > "Each item x is represented by a set ... The descriptive text ... of item i."
-  Several claims lack citations or evidence. For instance on Page 4:
  > "While vector quantization is commonly used for discretization, it proves unstable for training, leading to issues like item ID conflicts."
  This assertion is made without any supporting evidence.
-  In the ablation study on Page 7, the authors claim:
  > "Removing LLM-BS components such as DKI, GCT, and ATT (random index) achieves the worst results in different datasets, but still outperforms the vast majority of traditional baselines."
  It is unclear what remains of the framework when all key components are removed and how it still manages to outperform traditional baselines.

**Questions:**

Additional to the points mentioned above, I would like to ask the authors the following questions:
- How practical is the framework for real-world systems, especially in environments with limited computational resources or real-time constraints?
- How sensitive is the framework to the choice of clustering hyperparameters? Did you explore alternative clustering methods? Why was hierarchical K-Means chosen?
- Why is such an extreme compression ratio (~2,000,000:1) necessary, and how does it practically benefit the recommendation task?
- With each item represented by four tokens, how do you ensure that token collisions do not occur?
- How does the framework account for the inherently sequential nature of behavioral signals when clustering and discretizing them like static semantic signals?

**Reviewer Confidence:**

2: The reviewer is willing to defend the evaluation, but it is likely that the reviewer did not understand parts of the paper

**Scope:**

3: The work is somewhat relevant to the Web and to the track, and is of narrow interest to a sub-community

---

### Official Review · Reviewer_xmPF · 2024-12-02

**Novelty:** 4
**Technical Quality:** 6

**Review:**

LLM-BS is a decoder-only LLM-based generative recommendation framework that integrates endogenous and exogenous Behavioral and Semantic information in a non-intrusive manner. To overcome the drawback of LLM-based recommender, which is the lack of collaborative information learning, LLM-BS integrates user behavior into the semantic codebooks and introduces annealing adapter tuning where the sequential recommendation task is specifically tuned with LoRA after the pre-train phase with multiple tasks.

## Pros
1. The newly proposed codebook architecture and the training process is intuitively effective and reasonable for integrating more collaborative signals in LLM.
2. Clear writing style.
3. Significant improvement against new baselines including LC-Rec and LETTER, and convincing ablation study results to validate the effect of each proposed modules.

## Cons
1. Compared to other LLM-based recommenders with semantic IDs and collaborative knowledge, the novelty of the LLM-BS is rather limited. Previous work has already explored multiple implementations of the item encoding approaches, including RQ-VAE, hierarchical K-means, etc. The novelty (if the reviewer has correctly understood the model structure) mainly consist of splitting the codebook into content and behavioral part, and using LoRA to train on the target task one more time.
2. The authors may elaborate more on why previous LLM-based recommender does not incorporate collaborative signals enough, even when they utilize several auxiliary collaborative filtering tasks and hierachical codebook tokens.
3. See question 1.

**Questions:**

1. The reviewer does not quite get how the summary character $y_{[CON]}$ in Section 3.3.1 is supervised by any ground truth, and how it links with the predicted output tokens.

**Reviewer Confidence:**

4: The reviewer is certain that the evaluation is correct and very familiar with the relevant literature

**Scope:**

4: The work is relevant to the Web and to the track, and is of broad interest to the community

---

### Official Review · Reviewer_kxxk · 2024-12-02

**Novelty:** 3
**Technical Quality:** 5

**Review:**

**Summary**

The paper introduces LLM-BS, a method that represents items using semantic IDs embedded in prompts to fine-tune LLMs for recommendation tasks. Based on my understanding, its primary novelties compared to existing works like LC-Rec and LETTER are as follows: (1) item index learning via hierarchical clustering combined with a mix of textual and behavioral features; (2) the addition of a self-supervised loss for sequential recommendation tasks; and (3) a so-called annealing adapter tuning method, though it is insufficiently detailed in the paper, leaving the exact implementation unclear. The experiments were conducted on three public datasets.

**Strengths**
1. The paper studies an important task: learning semantic IDs for items and fine-tuning LLMs as recommendation models.
2. Experiments are conducted on three public datasets.

**Weaknesses**
1. Potentially lower baseline results:
The baseline results for the Instruments dataset in Table 2 appear to be directly copied from Table 3 of the LC-Rec paper for most baselines such as GRU4Rec, FDSA, and TIGER. However, the performance reported for LC-Rec is inconsistent. In the original LC-Rec paper, an NDCG@10 of 0.0926 is reported, which matches the result for LLM-BS. Yet, in this paper, the NDCG@10 for LC-Rec is listed as 0.890 without any explanation. If I missed something, please clarify.
2. Limited novelty:
    1. The proposed approach largely follows the framework of LC-Rec, which represents items using semantic IDs and fine-tunes LLMs with prompts containing these IDs. The fine-tuning instructions used are derived from LC-Rec. Beyond this, the novelty seems to lie in the construction of item indices, the contrastive loss, and the annealing adapter tuning, making the contributions kind of incremental.
    2. The idea of learning item indices from both textual and behavioral features is not new.
        * Conceptually, incorporating collaborative filtering (behavioral) signals for item index construction has been explored in prior work. For example:
            * GPTRec [1] quantizes SVD features to generate item indices.
            * P5-CID uses spectral clustering to generate indices.
        * Technically, hierarchical clustering has also been widely used for constructing semantic IDs in works like DSI [2] and P5-CID.
        * Combining textual and collaborative signals has been previously studied, e.g., LETTER uses collaborative filtering embeddings to regularize semantic ID learning, and P5-CID discusses several methods for integrating collaborative and textual/semantic IDs.
3. Marginal improvement from contrastive loss:
LLM-BS optimizes an additional contrastive learning objective between (1) the final hidden states of instructions and (2) the textual/behavioral features of target items. While the approach is interesting, it appears to yield only marginal performance improvements, as shown in Table 4, without statistically significant results.
4. Vague description of annealing adapter tuning:
Section 3.4.2 introduces annealing adapter tuning but lacks sufficient detail or clear formats to understand how it is applied. My assumption is that it might resemble modules used in LoRA fine-tuning.
5. Unavailability of code during the review phase:
The code is not publicly accessible, which limits reproducibility.

[1] Generative Sequential Recommendation with GPTRec.

[2] Transformer Memory as a Differentiable Search Index.

**Questions:**

1. In line 444, you mention that "256 * 4 tokens lead to 4 * 10^9 items," indicating "a capacity more than adequate for real-world applications." However, in line 324, you also claim that the proposed method minimizes the number of tokens. Could you clarify how the token usage is considered "minimal"?
2. In line 394, you state that "(quantization) proves unstable for training." Does this conclusion come from prior research? If so, please provide references.
3. In line 557, could you elaborate on what you mean by "high-grade sequence recommendation data"?
4. How were the LC-Rec results in Table 2 obtained?

**Reviewer Confidence:**

4: The reviewer is certain that the evaluation is correct and very familiar with the relevant literature

**Scope:**

4: The work is relevant to the Web and to the track, and is of broad interest to the community

---

### Official Review · Reviewer_bxFj · 2024-12-02

**Novelty:** 4
**Technical Quality:** 4

**Review:**

Pros:
1. This paper proposed an innovative decoder-only LLM-based generative recommendation framework that synergistically integrates endogenous and exogenous behavioral and semantic information

2. This paper also introduced the minimal number of tokens to efficiently represent a vast set of candidates and an  Annealing Adapter to optimize the balance between the model’s textual reasoning abilities and recommendation accuracy•

3. Extensive experiments across three public recommendation benchmarks demonstrate the superiority of LLM-BS

Cons:
1. The results show the proposed model archives better results than traditional and transformer-based models. However, the datasets used in this paper are quite sparse, which makes the comparison unfair to the traditional and transformer-based models. It would be better to use datasets with different sparsities to compare.

2. This paper designs an efficient indexing structure, however the number of users and items in the used datasets are around around 10 or 20 thousand. It would be better to test the proposed indexing structure with large-sized datasets to illustrate the effectiveness.

3. The paper claims they address the alignment between the collaborative information and LLM, but the baseline only compares the LC-Rec, more related methods would be better to compare with, such as very similar works: Large Language Models meet Collaborative Filtering: An Efficient All-round LLM-based Recommender System, RA-Rec: An Efficient ID Representation Alignment Framework for LLM-based Recommendation, Collaborative Large Language Model for Recommender Systems.

**Questions:**

Please refer the cons.

**Reviewer Confidence:**

4: The reviewer is certain that the evaluation is correct and very familiar with the relevant literature

**Scope:**

4: The work is relevant to the Web and to the track, and is of broad interest to the community